# H2ZIXY: Pauli spin matrix decomposition of real symmetric matrices

Rocco Monteiro Nunes Pesce[1] and Paul D. Stevenson[1*]

**1** Department of Physics, University of Surrey, Guildford, Surrey, GU2 7XH, UK
* p.stevenson@surrey.ac.uk

November 2, 2021

## Abstract

**We present a code in Python3 which takes a square real symmetric matrix, of arbitrary size, and decomposes it as a tensor product of Pauli spin matrices. The application to the decomposition of a Hamiltonian of relevance to nuclear physics for implementation on quantum computer is given.**

## 1   Introduction

In the field of quantum computing, practical quantum computers are often realised in terms of qubits – two level quantum system – which can be made to undergo a series of quantum logic gate operations. A natural mathematical way to express the operations is with the set of Pauli spin matrices, along with the identity matrix. For example, the variational quantum eigensolver [1] for a many-body system is often implemented by using the Jordan-Wigner transformation to turn a creation or annihilation operator into Pauli matrices [2, 3]. There are many ways to encode a Hamiltonian in terms of Pauli matrices [4, 5], and here we present a method to represent an N×N real symmetric matrix as the combination of Pauli matrices that composes the given matrix through Kronecker products. As a sample application, we apply the method to a Hamiltonian describing the deuteron using effective field theory as previously implemented on a quantum computer under the Jordan-Wigner mapping [6]. We note the wider interest of re-expressing matrices in a matter suitable for quantum computing [7] as well as the relevance of our approach to the field of tensor networks [8].

Including the Identity, $I$, the Pauli spin matrices can be written (in the usual $z$-diagonal basis) as

$$I = \begin{pmatrix} 1 & 0 \\ 0 & 1 \end{pmatrix}, \qquad X = \begin{pmatrix} 0 & 1 \\ 1 & 0 \end{pmatrix}, \qquad Y = \begin{pmatrix} 0 & -i \\ i & 0 \end{pmatrix}, \qquad Z = \begin{pmatrix} 1 & 0 \\ 0 & -1 \end{pmatrix}. \tag{1}$$

An arbitrary linear combination of the $I$, $X$ and $Z$ with real coefficients, is

$$a_0 I + a_1 X + a_3 Z = \begin{pmatrix} a_0 + a_3 & a_1 \\ a_1 & a_0 - a_3 \end{pmatrix}. \tag{2}$$

One can see that any real 2×2 symmetric matrix can be represented in this form, and the three matrices $I$, $X$, and $Z$ form a complete basis for real symmetric 2×2 matrices. Formally one can equate the matrix elements in (2) with the known values in the given matrix, and solve the resulting set of equations for the unknowns $a_0$, $a_1$ and $a_3$, though it is trivial to do so for the 2×2 case.

Extending to larger matrices, the Kronecker product of Pauli matrices form suitable complete bases, and the purpose of the present work is to take an arbitrary real symmetric N×N matrix and to give the representation in terms of Kronecker (or *tensor*) products of Pauli matrices. For a matrix which has a power of two order the decomposition is unique, since the Kronecker product of $N$ Pauli matrices is of order $2^N$. Matrices of any other order need to be padded to be of a power of two order, and the method of padding is decribed in the following section, along with the algorithm in general. In equating the higher order analogue of equation (2) with the given known matrix, a set of equations for the unknowns is yielded which can be solved by any suitable linear algebra package.

## 2   Algorithm

In this section the generic algoritm is given. A particular example is worked through in the following section.

The given input for this method is a user-supplied order $N$ square matrix, $H$.

*Step 1:* If $H$ is not of a power-of-two order, increase the size of $H$ and pad the new elements with zero. The decision to use zero is motivated by the application of this method to Hamiltonians of systems with bound (negative energy) states whose ground state energy is sought via a variational technique. Hence a choice of adding extra eigenvalues of zero will leave the variational minimum ground state unchanged. A user of the code may wish to revisit the choice of padding by putting eigenvalues of their choice in the diagonal elements of the padding region.

*Step 2:* From the set $\{IXYZ\}$, generate all permutations with $\log_2 N$ factors, with repetition allowed. Here $N$ is the order of the possibly enlarged $H$ matrix from step 1. Remove any permutations with an odd number of $Y$s since they will give an imaginary component. Generate the set of Kronecker products of the permutations, and store in a dictionary.

*Step 3:* Loop through the elements of the Hamiltonian H. Form an equation whose left hand side is the element of H and the right hand side is the sum of corresponding matrix elements of all Kronecker products generated in step 2 with an unknown coefficient in front. Encode the equation as one row of $M$ in the matrix-vector equation $Ma = h$ where $a$ is a vector of all the unknowns and $h$ is the Hamiltonian reshaped as a column vector

*Step 4:* Use routine from Python *numpy* package to solve set of equations for unknowns $\{a_n\}$ and construct a string representation of the matrix decomposition.

## 3   Usage guide

The software is supplied as a single Python3 function. We have not packaged it as part of a library since it was not written as such, and is not allied to any other functions.

The function `h2zixy` takes a single argument – a NumPy [9] array – and returns a text string containing the input matrix in Pauli matrix form.

It is expected that the user will take the function and embed in their own project as they find most useful. The distributed file `h2zixy.py` file includes an `if __name__ == '__main__'` clause to work out the small example detailed in the next section.

## 4 Worked example

As an example, we work through a case of a 3×3 matrix. We take the following matrix, which representes the energy of the deuteron in an oscillator basis restricted to three oscillator states [6]:

$$
H = \begin{pmatrix} -0.43658111 & -4.28660705 & 0 \\ -4.28660705 & 12.15 & -7.82623792 \\ 0 & -7.82623792 & 19.25 \end{pmatrix}.
\tag{3}
$$

This is extended to the next power of two order, i.e. 4×4 by padding with zeros::

$$
H = \begin{pmatrix} -0.43658111 & -4.28660705 & 0 & 0 \\ -4.28660705 & 12.15 & -7.82623792 & 0 \\ 0 & -7.82623792 & 19.25 & 0 \\ 0 & 0 & 0 & 0 \end{pmatrix}.
\tag{4}
$$

Since the order $4 = 2^2$ we make a list of all pairs of Pauli matrices which do not have an odd numbers of $Y$ matrices. This list is

$$
II, IX, IZ, XI, XX, XZ, YY, ZI, ZX, ZZ.
\tag{5}
$$

Each of the elements in the list has a 4×4 matrix representation made, which are (from intermediate values not usually printed in the code):

$$
II = \begin{pmatrix} 1 & 0 & 0 & 0 \\ 0 & 1 & 0 & 0 \\ 0 & 0 & 1 & 0 \\ 0 & 0 & 0 & 1 \end{pmatrix}, \quad
IX = \begin{pmatrix} 0 & 1 & 0 & 0 \\ 1 & 0 & 0 & 0 \\ 0 & 0 & 0 & 1 \\ 0 & 0 & 1 & 0 \end{pmatrix}, \quad
IZ = \begin{pmatrix} 1 & 0 & 0 & 0 \\ 0 & -1 & 0 & 0 \\ 0 & 0 & 1 & 0 \\ 0 & 0 & 0 & -1 \end{pmatrix},
$$

$$
XI = \begin{pmatrix} 0 & 0 & 1 & 0 \\ 0 & 0 & 0 & 1 \\ 1 & 0 & 0 & 0 \\ 0 & 1 & 0 & 0 \end{pmatrix}, \quad
XX = \begin{pmatrix} 0 & 0 & 0 & 1 \\ 0 & 0 & 1 & 0 \\ 0 & 1 & 0 & 0 \\ 1 & 0 & 0 & 0 \end{pmatrix}, \quad
XZ = \begin{pmatrix} 0 & 0 & 1 & 0 \\ 0 & 0 & 0 & -1 \\ 1 & 0 & 0 & 0 \\ 0 & -1 & 0 & 0 \end{pmatrix},
$$

$$
YY = \begin{pmatrix} 0 & 0 & 0 & -1 \\ 0 & 0 & 1 & 0 \\ 0 & 1 & 0 & 0 \\ -1 & 0 & 0 & 0 \end{pmatrix}, \quad
ZI = \begin{pmatrix} 1 & 0 & 0 & 0 \\ 0 & 1 & 0 & 0 \\ 0 & 0 & -1 & 0 \\ 0 & 0 & 0 & -1 \end{pmatrix}, \quad
ZX = \begin{pmatrix} 0 & 1 & 0 & 0 \\ 1 & 0 & 0 & 0 \\ 0 & 0 & 0 & -1 \\ 0 & 0 & -1 & 0 \end{pmatrix},
\tag{6}
$$

$$
ZZ = \begin{pmatrix} 1 & 0 & 0 & 0 \\ 0 & -1 & 0 & 0 \\ 0 & 0 & -1 & 0 \\ 0 & 0 & 0 & 1 \end{pmatrix}.
$$

Next, we loop over the ten unique matrix elements, with index $i, j$, of the matrix $H$ in (4) and for each element construct a row vector whose elements are the ten elements extracted from the same index $i, j$ of each of the matrices in (6) in the order given. So, for the first (top left) element, the row vector reads

$$
\begin{pmatrix} 1 & 0 & 1 & 0 & 0 & 0 & 0 & 1 & 0 & 1 \end{pmatrix}.
\tag{7}
$$

This row vector when multiplied by a column vector of coefficients for each of the Pauli matrix Kronecker products in (6) and equated to the top left element in $H$ (-0.43658111) gives one of

the simultaneous equations to be solved. By looping over all unique elements in $H$ ten equations for ten unknowns are constructed. The row vectors as in (7) are combined into a matrix, $M$. With the elements of $H$ packed into a 10 element column vector, $h$. Then the solution of $Ma = h$ for the coefficients $a$ gives the final answer – the coefficients of the Pauli Kronecker product representtion of the matrix $H$.

The full equation for our example is

$$
\begin{pmatrix}
1 & 0 & 1 & 0 & 0 & 0 & 0 & 1 & 0 & 1 \\
0 & 1 & 0 & 0 & 0 & 0 & 0 & 0 & 1 & 0 \\
0 & 0 & 0 & 1 & 0 & 1 & 0 & 0 & 0 & 0 \\
0 & 0 & 0 & 0 & 1 & 0 & -1 & 0 & 0 & 0 \\
1 & 0 & -1 & 0 & 0 & 0 & 0 & 1 & 0 & -1 \\
0 & 0 & 0 & 0 & 1 & 0 & 1 & 0 & 0 & 0 \\
0 & 0 & 0 & 1 & 0 & -1 & 0 & 0 & 0 & 0 \\
1 & 0 & 1 & 0 & 0 & 0 & 0 & -1 & 0 & -1 \\
0 & 1 & 0 & 0 & 0 & 0 & 0 & 0 & -1 & 0 \\
1 & 0 & -1 & 0 & 0 & 0 & 0 & -1 & 0 & 1
\end{pmatrix}
\begin{pmatrix}
a_{II} \\ a_{IX} \\ a_{IZ} \\ a_{XI} \\ a_{XX} \\ a_{XZ} \\ a_{YY} \\ a_{ZI} \\ a_{ZX} \\ a_{ZZ}
\end{pmatrix}
=
\begin{pmatrix}
-0.43658111 \\ -4.28660705 \\ 0 \\ 0 \\ 12.15 \\ -7.82623792 \\ 0 \\ 19.25 \\ 0 \\ 0
\end{pmatrix}
\tag{8}
$$

The solution, using the NumPy [9] routine `linalg.solve` gives (with coefficients written to 4 significant figures)

$$H = 7.766II - 2.143IX + 1.641IZ - 3.913XX - 3.913YY - 1.859ZI - 2.143ZX - 7.984ZZ \tag{9}$$

## 5  Conclusion

The Python function described in this paper performs the task of decomposing a square matrix into Kronecker products of Pauli spin matrices is a systematic and straightforward way. It performs its role well for cases of interest to the authors, but other users may benefit from small changes, such as allowing padding with non-zero dummy eigenvalues; numerical output of coefficients in an array; speedup in cases of large N; exception handling; input checking. We hope the presented form provides a sufficient solution for some, and a helpful starting point for others.

## Acknowledgements

Useful discussions with Isaac Hobday and James Benstead are acknowledged.

**Author contributions**   The project was defined and overseen by PDS. RMNP performed the coding.

**Funding information**   RMNP was funded by EPSRC for a summer internship during which this project was conducted. PDS is funded by an AWE William Penney Fellowship

# A   Code

In the final version of the paper, the code will be linked to a repository associated with the journal. For the preprint version, we include a code listing here. The LaTeX source inlcudes the code in plain text form.

```python
def h2zixy(hamiltonian):
    """Decompose square real symmetric matrix into Pauli spin matrices

    argument:
    hamiltonian -- a square numpy real symmetric numpy array

    returns:
    a string consisting of terms each of which has a numerical coefficient
    multiplying a Kronecker (tensor) product of Pauli spin matrices
    """

    import itertools
    import numpy as np

    # coefficients smaller than eps are taken to be zero
    eps = 1.e-5

    dim = len(hamiltonian)

    # Step 1:expand Hamiltonian to have leading dimension = power of 2 and pad
    # with zeros if necessary

    NextPowTwo = int(2**np.ceil(np.log(dim)/np.log(2)))
    if NextPowTwo != dim:
        diff = NextPowTwo - dim
        hamiltonian = np.hstack((hamiltonian,np.zeros((dim, diff))))
        dim = NextPowTwo
        hamiltonian = np.vstack((hamiltonian,np.zeros((diff,dim))))

    # Step 2: Generate all tensor products of the appropriate length with
    # all combinations of I,X,Y,Z, excluding those with an odd number of Y
    # matrices

    # Pauli is a dictionary with the four basis 2x2 Pauli matrices
    Pauli = {'I' : np.array([[1,0],[0,1]]),
             'X': np.array([[0,1],[1,0]]),
             'Y': np.array([[0,-1j],[1j,0]]),
             'Z': np.array([[1,0],[0,-1]])}

    NumTensorRepetitions = int(np.log(dim)/np.log(2))
    NumTotalTensors = 4**NumTensorRepetitions
    PauliKeyList = []
    KeysToDelete = []
    PauliDict = {}

    def PauliDictValues(l):
        yield from itertools.product(*([l] * NumTensorRepetitions))

    #Generate list of tensor products with all combinations of Pauli
    # matrices i.e. 'III', 'IIX', 'IIY', etc.
    for x in PauliDictValues('IXYZ'):
        PauliKeyList.append(''.join(x))

    for y in PauliKeyList:
        PauliDict[y] = 0

    for key in PauliDict:
        TempList = []
        PauliTensors = []
```

```
60          NumYs= key.count('Y')
61          TempKey = str(key)
62
63          if (NumYs % 2) == 0:
64              for string in TempKey:
65                  TempList.append(string)
66
67              for SpinMatrix in TempList:
68                  PauliTensors.append(Pauli[SpinMatrix])
69              PauliDict[key] = PauliTensors
70
71              CurrentMatrix = PauliDict[key].copy()
72
73              # Compute Tensor Product between I, X, Y, Z matrices
74              for k in range(1, NumTensorRepetitions):
75                  TemporaryDict = np.kron(CurrentMatrix[k-1], CurrentMatrix[k])
76                  CurrentMatrix[k] = TemporaryDict
77
78              PauliDict[key] = CurrentMatrix[-1]
79
80          else:
81              KeysToDelete.append(key)
82
83      for val in KeysToDelete:
84          PauliDict.pop(val)
85
86      # Step 3:  Loop through all the elements of the Hamiltonian matrix
87      # and identify which pauli matrix combinations contribute;
88      # Generate a matrix of simultaneous equations that need to be solved.
89      # NB upper triangle of hamiltonian array is used
90
91      VecHamElements = np.zeros(int((dim**2+dim)/2))
92      h = 0
93      for i in range(0,dim):
94          for j in range(i,dim):
95              arr = []
96              VecHamElements[h] = hamiltonian[i,j]
97              for key in PauliDict:
98                  TempVar = PauliDict[key]
99                  arr.append(TempVar[i,j].real)
100
101             if i == 0 and j == 0:
102                 FinalMat = np.array(arr.copy())
103
104             else:
105                 FinalMat = np.vstack((FinalMat, arr))
106
107             h += 1
108
109     # Step 4: Use numpy.linalg.solve to solve the simultaneous equations
110     # and return the coefficients of the Pauli tensor products.
111
112     x = np.linalg.solve(FinalMat,VecHamElements)
113     a = []
114     var_list = list(PauliDict.keys())
115
116     for i in range(len(PauliDict)):
117         b = x[i]
118         if abs(b)>eps:
119             a.append(str(b)+'*'+str(var_list[i])+'\n')
120
121     # Output the final Pauli Decomposition of the Hamiltonian
122     DecomposedHam = ''.join(a)
123     return DecomposedHam
124
125 if __name__ == '__main__':
```

```
126    import numpy as np
127
128    # for a sample calculation, take the Hamiltonian from the paper by
129    # Dumitrescu et al. (Phys. Rev. Lett. 120, 210501 (2018))
130
131    N = 20
132    hw = 7.0
133    v0 = -5.68658111
134
135    ham = np.zeros((N,N))
136    ham[0,0] = v0
137    for n in range (0,N):
138        for na in range(0,N):
139            if(n==na):
140                ham[n,na] += hw/2.0*(2*n+1.5)
141            if(n==na+1):
142                ham[n,na] -= hw/2.0*np.sqrt(n*(n+0.5))
143            if(n==na-1):
144                ham[n,na] -= hw/2.0*np.sqrt((n+1.0)*(n+1.5))
145
146    out= h2zixy(ham)
147    print(out)
```

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
