# Peer review of "H2ZIXY: Pauli spin matrix decomposition of real symmetric matrices"

_SciPost Physics Codebases_

## Round 1 · Referee Report · Anonymous (Referee 1) · 2021-12-1

Strengths

I am unable to find any strong points on the code and/or manuscript.

Weaknesses

1- Attempts to solve a well understood problem by a brute-force method. 2- The method proposed is not scalable; impossible to use even for small number of qubits. 3- The code does not provide any new insight or remarkable feature to speed up the problem. 4- The code does not come with proper documentation and testing, and therefore can not be considered a package.

Report

The code and complementary manuscript "H2ZIXY: Pauli spin matrix decomposition of real symmetric matrices" describe a function to find the Pauli matrix decomposition of a real-symmetric matrix. The Pauli decomposition is done by brute-force search of the 4^N Pauli operators of an N-qubit Hamiltonian.

I am unable to accept the code and attached manuscript for publication in SciPost codebases because it does not match any of the acceptance criteria described in https://scipost.org/SciPostPhysCodeb/about#criteria.

Addressing point-by-point the acceptance criteria: 1- There exist plenty of algorithms and packages that perform the Pauli decomposition of Hermitian matrices, thus the code does not address a need of the community. 2- The user guide is non-existent, only the comments on the code. The comments in the code do not serve as a guide, nor explain its and usability. The code is based on brut-force search and linear-equation solver, therefore there is no new insight on how to solve this problem. 3- The authors provide an example of the code usage. However, a quick test on 6 qubits takes a large amount of time and resources on a laptop. Scalability and usability are of essence for a code that aims at be widely used. 4- The code lacks of testing, benchmarking and comparison to other packages/methods for the same problem.

Requested changes

1- The code must first provide a new algorithm/method to find the Pauli decomposition of any real or imaginary Hermitian matrix without exponential cost. 2- The authors must search existing packages as a benchmarking and comparison of their method. 3- Test of the code are necessary for the code quality.

  • validity: poor
  • significance: poor
  • originality: low
  • clarity: ok
  • formatting: below threshold
  • grammar: below threshold

---

## Editorial Decision

awaiting_resubmission